# Dependence of Freeze-Out Parameters on Collision Energies and Cross-Sections

Muhammad Waqas [1,*], Atef AbdelKader [2,3], Muhammad Ajaz [4], Abdel Nasser Tawfik [5], Zafar Wazir [6], Abd Al Karim Haj Ismail [2,3,*], Shi Jun Luo [1,7] and Hafsa Zar Khan [8]

1 School of Mathematics, Physics and Optoelectronic Engineering, Hubei University of Automotive Technology, Shiyan 442002, China
2 College of Humanities and Sciences, Ajman University, Ajman P.O. Box 346, United Arab Emirates
3 Nonlinear Dynamics Research Center (NDRC), Ajman University, Ajman P.O. Box 346, United Arab Emirates
4 Department of Physics, Abdul Wali Khan University Mardan, Mardan 23200, Pakistan
5 Faculty of Engineering, Future University in Egypt (FUE), Fifth Settlement, New Cairo 11835, Egypt
6 Department of Physics, Ghazi University, Dera Ghazi Khan 32200, Pakistan
7 Collaborative Center for Optoelectronic Technology, Hubei University of Automotive Technology, Shiyan 442002, China
8 Center for High Energy Physics, Punjab University, Lahore P.O. Box 54590, Pakistan
* Correspondence: waqas_phy313@yahoo.com or 20220073@huat.edu.cn (M.W.);
a.hajismail@ajman.ac.ae (A.A.K.H.I.)

**Abstract:** We analyzed the transverse momentum spectra ($p_T$) reported by the NA61/SHINE and NA49 experiments in inelastic proton–proton ($pp$) and central Lead–Lead ($Pb - Pb$), Argon–Scandium ($Ar - Sc$), and Beryllium–Beryllium ($Be - Be$) collisions with the Blast-wave model with Boltzmann–Gibbs (BWBG) statistics. The BGBW model was in good agreement with the experimental data. We were able to extract the transverse flow velocity ($\beta_T$), the kinetic freeze-out temperature ($T_0$), and the kinetic freeze-out volume ($V$) from the $p_T$ spectra using the BGBW model. Furthermore, we also obtained the initial temperature ($T_i$) and the mean transverse momentum ($<p_T>$) by the alternative method. We observed that $T_0$ increases with increasing collision energy and collision cross-section, representing the colliding system's size. The transverse flow velocity was observed to remain invariant with increasing collision energy, while it showed a random change with different collision cross-sections. In the same way, the kinetic freeze-out volume and mean transverse momentum increased with an increase in collision energy or collision cross-section. The same behavior was also seen in the freeze-out temperature, which increased with increasing collision cross-sections. At chemical freeze-out, we also determined both the chemical potential and temperature and compared these with the hadron resonance gas model (HRG) and different experimental data. We report that there is an excellent agreement with the HRG model and various experiments, which reveals the ability of the fit function to manifest features of the chemical freeze-out.

**Keywords:** transverse momentum spectra; kinetic freeze-out temperature; transverse flow velocity; kinetic freeze-out volume; initial temperature; collision cross-section

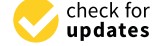

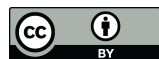

## 1. Introduction

In the early stages of nuclear collisions, a new state of matter, quark–gluon plasma (QGP), is likely to be formed under the extreme conditions of temperatures and densities. This matter has a lifetime of 7–10 fm/c, after which it changes quickly to a system of hadron gas. Due to multi-partonic interactions in the collision cross-section, information about the initial condition of the system is lost, such as the transverse excitation degree and the dynamic expansion of the collision cross-section. We can determine the final state behavior of such systems based on the number of particles produced and their energy and transverse momentum spectra.

During the evolution of heavy-ion collisions, the constituents of hot and dense matter interact with each other elastically or in-elastically and evolve into a new state of matter. This phenomenon of particle decoupling is called freeze-out, of which there are two kinds. Due to the expansion of the system, the inelastic collisions stop and the mean free path for the interactions becomes comparable to the size of the system. In addition, the abundances of different particle species become fixed. This is referred to as chemical freeze-out [1–3]. The chemical freeze-out stage is followed by the second kind. Although the relative fractions of the particles are fixed, at the chemical freeze-out stage, they continue to interact with each other until the final-state interactions between the particles are no longer effective. This is called the kinetic freeze-out where all interactions stop, and the transverse momentum spectra of the produced particles do not change. Therefore, the transverse momentum spectra of the particles are very important because they contain the necessary information about the final-state particles, including $T_0$, $(\beta_T)$, $(V)$, and the time of travel of the particles. At the chemical freeze-out stage, the chemical freeze-out temperature ($T_{ch}$) and baryon chemical potential ($\mu_b$) can be determined based on well-defined thermodynamic conditions [3,4]. Besides $T_0$ and $T_{ch}$, $T_i$ is also an important quantity because it determines the evolution of the entire colliding system.

Not only the systems with large collision cross-sections are conjectured to form QGP, small systems (especially with high multiplicity) are also expected to have such an effect. This is due to their small volumes of violent collision regions. At the top RHIC and LHC energies, pp collisions with high multiplicity show similar behavior to peripheral AA collisions, while at lower energies, from a few GeV to 10 GeV, the situation is different due to the dominance of baryons [5].

In the present work, we use the Blast-wave model with Boltzmann–Gibbs statistics [3,6–8] to analyze the transverse momentum spectra of the pions in various systems with different collision cross-sections at different energies and extract the kinetic freeze-out temperature, transverse flow velocity, and kinetic freeze-out volume. We will show the dependence of the above parameters on collision energy and collision cross-section. In the recent literature [3,7,9–23], the dependence of these parameters are in contradiction to one another. The contradiction involves the dependence of $T_0$ on the collision cross-section, which could increase, decrease, or be invariant from the central to the peripheral collisions (with increasing energy, especially for RHIC beam energies after 39 GeV). It is also worth mentioning that different models have been used by different groups and that the models are used under different conditions and constraints, leading to different results, as in [12]. In addition, there is a contradiction about the decoupling of the particles, which follow the single, double, triple, or multiple freeze-out scenarios, as in some cases, the particles' decoupling is reported to be affected by the coalescence and iso-spin symmetry. In fact, it is always very useful to conduct more studies on these topics and try to finalize some corroborative conclusions. Moreover, we will also extract $T_i$, $T_{ch}$, and $T_0$. Furthermore, the $\langle p_T \rangle$ shall be extracted. This allows us to study their dependence on collision energy and cross-section.

Before going to the next section, we would like to point out that we chose the Blast-wave model with Boltzmann–Gibbs statistics in the present work because it is the most direct and simpler model with fewer parameters, as well as being closer to the ideal gas model [24–28]. In addition, the performance of the BGBW model is good in the low $p_T$ region ($p_T$ = 0–3.4 or 4 GeV/c) and meets the needs of this research, and the reason behind the selection of pions is that the temperature obtained from their spectra is closest to the source temperature. The rest of this paper is organized as follows: The methods and models are discussed in Section 2. The results and discussions are presented in Section 3. The summary and conclusion are described in Section 4.

## 2. Methods and Models

In high-energy collisions, the $p_T$ spectra of the produced particles are very complex since the function describing $p_T$ can have different forms. The complex $p_T$ structure refers

to different $p_T$ regions, which correspond to different interaction mechanisms and have different properties. These regions include the $p_T < 4$–$6$ GeV/c as the first region, followed by the second region in the energy range from $p_T > 4$–$6$ GeV/c to $p_T < 20$ GeV/c, and the third region is $p_T > 20$ GeV/c. The effects and changes by the medium appear in the first $p_T$ region, while they are weak in the second $p_T$ range, while the third $p_T$ region shows the negligible influence of the medium given by the nuclear transparency. In addition, it is not enough to use a single probability density function to describe the $p_T$ spectra [29]. There are various $p_T$ regions [30] according to the model analysis that is described in our previous work [31]. Soft excitation and hard scattering are the two main processes for particle production. The soft excitation process results in the production of most light flavor particles, whose $p_T$ range is narrow (less than 2–3 GeV/c), while the hard scattering process exists in a wide $p_T$ range ($p_T > 3$ GeV/c); in addition, some light flavor particles are also produced in this process. In some cases of not-too-high collision energies, the hard scattering process can be underestimated, and soft excitation plays the main role in particle production. In general, due to the small fraction of the hard process in the narrow $p_T$ range, the hard scattering process does not contribute to temperature or flow velocity. For the soft excitation process, we have various choices of formalism, which include, but are not limited to, the standard distribution [32], Tsallis statistics [33–36], Erlang distribution [37–39], the Schwinger mechanism [40–43], the Blast-wave model with Boltzmann–Gibbs statistics [6,44], the Blast-wave model with Tsallis statistics [45–47], and Hagedorn thermal model distribution [48].

In the present work, we use the Blast-wave model with Boltzmann–Gibbs statistics, which assumes that the particles are locally thermalized at the thermal/kinetic freeze-out temperature and that they are moving with common transverse collective flow velocities [5,49]. Let us assume a thermal source, which is radially boosted, has $T_0$ and $\beta_T$, and the $p_T$ spectra distribution of the particles is given as:

$$
\begin{aligned}
f(p_T) &= \frac{1}{N}\frac{dN}{dp_T} = \frac{1}{N}\frac{gV}{(2\pi)^2}p_T m_T \int_0^R r\,dr \\
&\times\ I_0\left[\frac{p_T \sinh(\rho)}{T_0}\right]K_1\left[\frac{m_T \cosh(\rho)}{T_0}\right],
\end{aligned}
\tag{1}
$$

where $m_T$ ($m_T = \sqrt{p_T^2 + m_0^2}$) represents the transverse mass of the particles, and $g$ is the spin degeneracy factor of the particle ($g = 1$ for pions, for example). $I_0$ and $K_1$ are the modified Bessel function of the first and second kind, respectively. The radial flow velocity profile, $\rho$, is defined as $\tanh^{-1}[\beta(r)]$, where $r/R$ represents the relative radial position in thermal source. The average $\beta(r)$ can be obtained from $\langle\beta_T\rangle=2\beta_S/(n_0+2)$, where $n_0$ is the self similar flow velocity profile, and its value can be 1 [5] or 2 [50], or it may also be considered as a free parameter [49]. In some cases, it is possible that BGBW does not fit the whole $p_T$ region, then we use the two-component model [17,31].

In the fit process, the extracted parameters usually have a correlation, such as $T_0$, which becomes larger in some cases, and a smaller $\beta_T$ can lead to similar results if a smaller $T_0$ and larger $\beta_T$ are used. This is due to the influence of the $p_T$ range and also $n_0$ if taken as a free parameter. To reduce the effect of such a correlation, we need to analyze the $\langle p_T \rangle$ and the root-mean-square $p_T$ over $\sqrt{2}(\sqrt{\langle p_T^2\rangle/2})$. We can calculate $\langle p_T \rangle$ and $\sqrt{\langle p_T^2\rangle/2}$ from the fit function over a given $p_T$ range, where $\sqrt{\langle p_T^2\rangle/2}$ is the initial temperature of the interacting system according to the string percolation model [51–53].

## 3. Results and Discussion

The transverse momentum ($p_T$) spectra of $\pi^-$ mesons produced in inelastic (INEL) proton–proton (pp), Beryllium–Beryllium (Be–Be), Argon–Scandium (Ar–Sc), and Lead–Lead (Pb–Pb) collisions, at different energies, are presented in Figure 1 (top panels (a) and (b) show the $p_T$ spectra of the pions in inelastic (INEL) pp collisions and the most central Be–

Be, Ar–Sc, and Pb–Pb collisions, respectively, at $|y| = 0.1$ rapidity). The symbols represent the experimental data of the NA61/SHINE [54–56] and NA49 experiments [57,58] measured at SPS CERN. The curve represents the results of our fit by the Blast-wave model with Boltzmann–Gibbs statistics. One can see that Equation (1) can describe the experimental data well, and the related parameters are extracted from the fit of Equation (1) to the experimental data following the least-squares method. We checked the stability of the obtained parameter values by repeating the fit procedures three times and changing the initial values of the parameters. As a result, the stability of the obtained parameters was fully confirmed (they stayed practically the same). The data in Figure 1 (panel (a), (b), and (c)) are taken from ref. [54–56], respectively, while the data in Figure 1 (panel (d)) are taken from ref. [57,58]. To see the fit results clearly, the spectra of the pions in pp collisions at 40, 80, and 158 A GeV/c in panel (a) are scaled by 4, 14, and 40, respectively. In panel (b), at 30, 40, and 150 A GeV/c, the spectra are scaled by 4, 2, and 1/8, respectively. In panel (c), at 40, 80, and 160 A GeV/c, the spectra are scaled by 20, 8, and 4, respectively. Furthermore, we would like to mention that in Figure 1, we are using the hydrodynamical model to fit the experimental spectra, which is not available at these energies, which results in few data points at high energies. However, if there are more data points in the same pt range, it will improve the quality of the fit but will not have a large effect on the final results. Moreover, if the $p_T$ range is to be changed, this might have an effect on the results.

The lower layer in each panel represents the corresponding ratio of data/fit. The related values of free parameters and $\chi^2$ and the degrees of freedom (*dof*) are presented in Table 1. One can see that Equation (1) provides an approximate well fit to the data in all collisions at all energies. In Table 1, it should be noted that different rapidity could be used from [54] for pp collisions, where the data points are comparatively more, but we only used the rapidity $|y| = 0.1$ in the present work because the data for pp, Be–Be, Ar–Sc, and Pb–Pb collisions for another similar rapidity are not available. The change in rapidity will have an effect on the collective parameters [59], while the present work is focused on the dependence of the collective parameters on collision energies and collisions cross-sections. In addition, $N_0$, used in the table, is a normalization constant that compares the fit function with the experimental spectra.

In the present work, the data/fit ratio is in the acceptable range. In some cases, especially in the lower panels of Figure 1, the data/fit ratio is slightly larger than 2. The larger tension of the data to the curve from unity is in the very low $p_T$ range of $p_T < 0.5$ GeV/c. The reason for this is that $pT < 0.5$ GeV/c is the very soft region where a large fraction of pions originate from resonance decay, which is not covered by the model. It is true that if the data/fit ratio is close to unity, this will give us the confidence that the model used can describe the data very well, but this is mostly possible in the case that we use the two components of the model. However, the contribution of the second component of the model to the parameters is very little; therefore, we do not consider using it in the present work. To study the change in the trend of the parameters, Figure 2 shows the dependencies of the kinetic freeze-out temperature, transverse flow velocity, and kinetic freeze-out volume on collision energy and collision cross-section. The panels (a), (b), and (c) show the result of the $T_0$, $\beta_T$, and $V$, respectively. The different symbols represent different systems. The trend of the parameters from left to right shows the energy dependence of the corresponding parameter, while from up to down, their dependence on collision cross-section is displayed. In panel (a), the dependence of the kinetic freeze-out temperature on the collision energy and collision cross-section is shown. We can observe that as the collision energy increases, $T_0$ also increases. The reason behind this is that at high energies, the collision is very violent, which gives a higher excitation to the system. The higher the energies, the higher the degree of excitation the system will obtain. Furthermore, it can also be observed that $T_0$ in pp collisions is less than in the other three collisions, while in Pb–Pb, it is the largest, followed by Ar–Sc and then Be–Be collisions. This indicates that $T_0$ depends on the collision cross-section interaction. The larger the collision cross-section, the higher the $T_0$ we observe. The proton–proton system has a smaller cross-section, while

the Lead–Lead system has the largest collision cross-section and, thus, the largest $T_0$ value. This is in agreement with the results in [13]. Panel (b) shows the dependence of $\beta_T$ on the energy and collision cross-section. At present, we observe that $\beta_T$ remains unchanged with increasing collision energy. We believe that this is due to the reason that collective behavior does not change with increasing energy. In addition, $\beta_T$ changes randomly for every system, as reported in the literature [60,61]. In the present case, no dependence of $\beta_T$ can be observed on the collision cross-section. $\beta_T$ is normally taught to be a decreasing function with increasing cross-section because denser systems should have lower flow, but at the same time, higher pressure in the system tends to increase it. Therefore, there is no dependence of $\beta_T$ observed on the collision cross-section. In panel (c), the dependence of $V$ on the collision energy and the collision cross-section is presented. One can see that $V$ increases as the collision energy increases. The reason behind this is that there is a larger initial bulk system at high energies. The increase in energy results in a long evolution time, which corresponds to a larger partonic system, and $V$ becomes larger in a large partonic system. Furthermore, we can also observe that $V$ is larger for Pb–Pb collisions, followed by Ar–Sc and then Be–Be collisions, and it is the lowest in pp collisions. This indicates its dependence on the collision cross-section (the size of the interacting system).

**Table 1.** List of the parameters. The "-" sign is used in some places instead of dof. In fact, it is not the fit result. If dof < 0, we use "-" instead of negative values.)

| Collisions | Energy | $T_0$ (GeV) | $\beta_T$ (c) | $V$ (fm³) | $N_0$ | $\chi^2$/dof |
|---|---|---|---|---|---|---|
| Figure 1a | 20 GeV | $0.085 \pm 0.004$ | $0.300 \pm 0.010$ | $1000 \pm 110$ | $0.27 \pm 0.04$ | 2/3 |
| pp | 31 GeV | $0.092 \pm 0.005$ | $0.302 \pm 0.009$ | $1200 \pm 110$ | $1.4 \pm 0.3$ | 0.2/- |
| | 40 GeV | $0.097 \pm 0.005$ | $0.308 \pm 0.008$ | $1340 \pm 102$ | $1.9 \pm 0.3$ | 1/- |
| | 80 GeV | $0.103 \pm 0.006$ | $0.304 \pm 0.008$ | $1513 \pm 100$ | $2 \pm 0.4$ | 0.03/- |
| | 158 GeV | $0.108 \pm 0.005$ | $0.308 \pm 0.011$ | $1600 \pm 95$ | $2 \pm 0.4$ | 0.1/1 |
| Figure 1b | 19 GeV | $0.100 \pm 0.006$ | $0.335 \pm 0.007$ | $1400 \pm 100$ | $0.0062 \pm 0.0004$ | 1/13 |
| Be–Be | 31 GeV | $0.107 \pm 0.006$ | $0.334 \pm 0.008$ | $1500 \pm 120$ | $0.0063 \pm 0.0003$ | 5/14 |
| | 40 GeV | $0.114 \pm 0.005$ | $0.335 \pm 0.008$ | $1645 \pm 110$ | $0.000075 \pm 0.000004$ | 3/14 |
| | 75 GeV | $0.120 \pm 0.006$ | $0.336 \pm 0.009$ | $1900 \pm 108$ | $8.5 \times 10^{-6} \pm 5 \times 10^{-7}$ | 4/14 |
| | 150 GeV | $0.125 \pm 0.006$ | $0.340 \pm 0.007$ | $2100 \pm 150$ | $8.9 \times 10^{-6} \pm 4 \times 10^{-7}$ | 1.5/14 |
| Figure 1c | 13 GeV | $0.104 \pm 0.004$ | $0.250 \pm 0.010$ | $2100 \pm 127$ | $1.5 \times 10^{-4} \pm 4 \times 10^{-5}$ | 13/16 |
| Ar–Sc | 19 GeV | $0.115 \pm 0.005$ | $0.240 \pm 0.009$ | $2200 \pm 130$ | $0.00154 \pm 0.0004$ | 11/16 |
| | 30 GeV | $0.123 \pm 0.006$ | $0.252 \pm 0.008$ | $2320 \pm 120$ | $0.019 \pm 0.004$ | 8/16 |
| | 40 GeV | $0.130 \pm 0.005$ | $0.240 \pm 0.008$ | $2500 \pm 119$ | $0.18 \pm 0.03$ | 12/16 |
| | 75 GeV | $0.136 \pm 0.005$ | $0.240 \pm 0.009$ | $2700 \pm 120$ | $2 \pm 0.3$ | 37/16 |
| | 150 GeV | $0.141 \pm 0.004$ | $0.180 \pm 0.008$ | $2900 \pm 130$ | $20 \pm 4$ | 12/16 |
| Figure 1d | 20 GeV | $0.133 \pm 0.005$ | $0.140 \pm 0.007$ | $2900 \pm 140$ | $55 \pm 8$ | 45/12 |
| Pb–Pb | 30 GeV | $0.139 \pm 0.005$ | $0.145 \pm 0.007$ | $3100 \pm 152$ | $300 \pm 32$ | 36/12 |
| | 40 GeV | $0.145 \pm 0.005$ | $0.128 \pm 0.007$ | $3274 \pm 147$ | $58 \pm 9$ | 23/10 |
| | 80 GeV | $0.157 \pm 0.004$ | $0.130 \pm 0.008$ | $3400 \pm 138$ | $770 \pm 100$ | 12/10 |
| | 160 GeV | $0.163 \pm 0.006$ | $0.130 \pm 0.008$ | $3700 \pm 143$ | $9000 \pm 800$ | 15/10 |

Figure 3a shows the dependence of $\langle p_T \rangle$ and $T_i$ on collision energy and collision cross-section. One can see that $\langle p_T \rangle$ increases with increasing collision energy and also in the collision cross-section because larger momentum (energy) is transferred at higher energies and large collision cross-sections, which results in further multiple scattering. However, in Figure 3b, the dependence of $T_i$ on energy and collision cross-section is presented. One can see that $T_i$ increases with increasing collision energy and is larger for a large collision cross-section system.

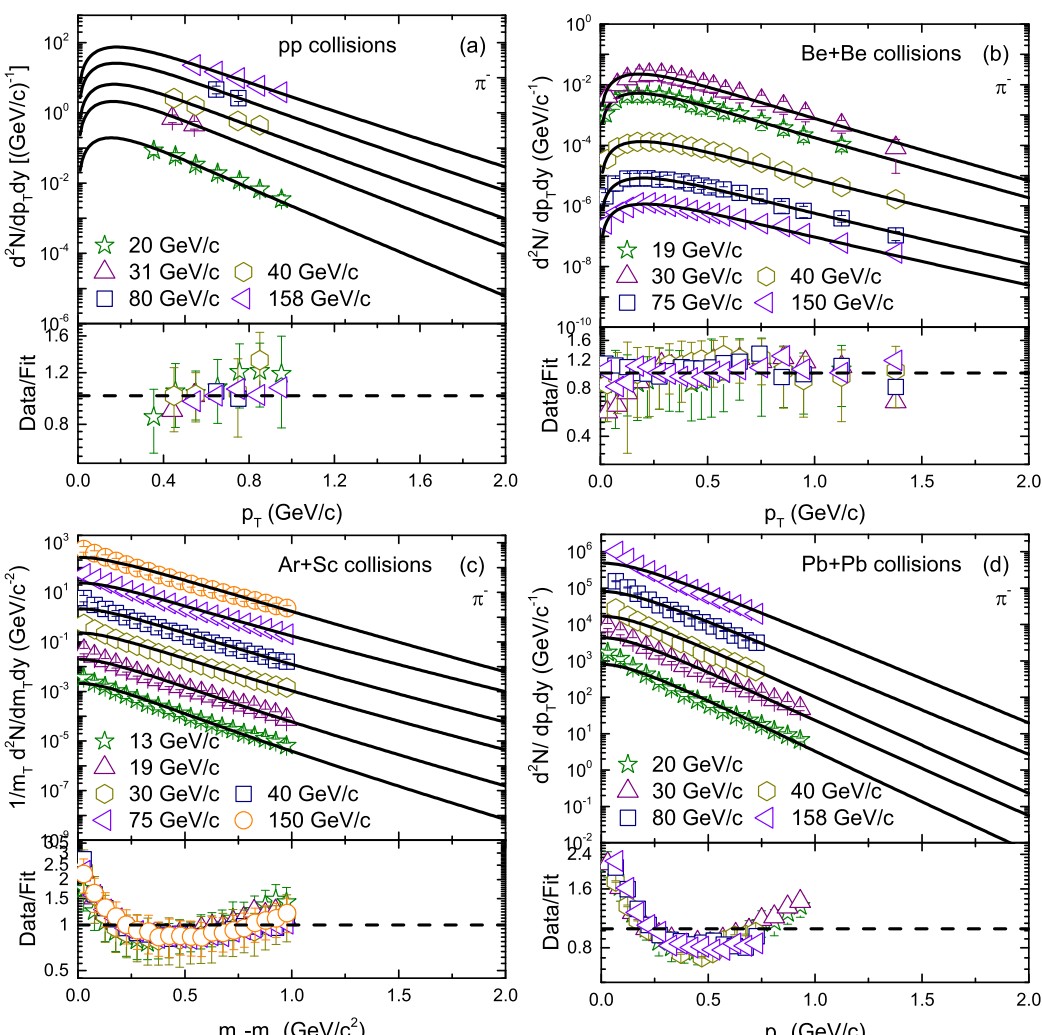

**Figure 1.** Transverse momentum spectra of $\pi^-$ produced in pp, and the most central Be−Be, Ar−Sc, and Pb−Pb collisions at different energies at $|y| = 0.1$ rapidity interval. The symbols are the experimental data of the NA61/SHINE [54–56] and NA49 experiments [54–56,58] measured by SPS CERN. The curves represent our fit by Equation (1). The corresponding data/fit ratios are followed in each panel.

We observe that $T_i$ is larger than $T_0$. Generally, $T_i$ is larger than the effective temperature ($T_{eff}$), and $T_{eff}$, in turn, is larger than $T_0$ because $T_{eff}$ includes the flow effect.

The other kind of freeze-out parameters—the temperature and baryon chemical potential, at chemical freeze-out—could be parameterized by various thermodynamics conditions, such as constant entropy density normalized to temperature cubed, constant higher-order moments of particle multiplicities, and constant trace anomaly (interaction rate) [62–64]. The present study suggests that

$$T_{ch} = \frac{T_{\lim}}{1 + \exp[2.60 - \ln(\sqrt{s_{NN}})/0.45]}, \tag{2}$$

$$\mu_b = \frac{1.303}{1 + 0.286\sqrt{s_{NN}}}, \tag{3}$$

where $T_{\lim} = 0.158$ GeV and $\sqrt{s_{NN}}$ are in the units of GeV [65]. The chemical freeze-out temperature is conjectured to lay between $T_i$ and $T_0$, generally. It is slightly larger than or approximately equal to $T_{eff}$. Such an order is in agreement with the order of time evolution of the interacting system. Figure 4 depicts $T_{ch}$ vs. $\mu_b$ and compares these results with the

experimental data and the hadron resonance gas model (HRG) [24]. With the experimental results, we mean $T_{ch}$ and $\mu_b$ in thermal models, such as the HRG model, fulfilling certain thermodynamic conditions and best reproducing the measurement of many particle ratios at various energies. Focusing on particle ratios largely eliminates the volume of the statistical system. The curve label "This work" is the result determined by the statistical hadronization model fits. The thin curve represents the results obtained by the HRG model under the condition of constant higher-order moments of particle multiplicities [62,63,66]. It should be mentioned that in creating this graph, we have included calculations across a very wide range of energies, exceeding the NA61/SHINE and NA49 energies. The latter is limited to a short range of $T_{ch}$ and $\mu_b$. The excellent agreement with both the HRG model and various experiments reveals the ability of the Blast-wave model with Boltzmann–Gibbs statistics to manifest the features of the chemical freeze-out.

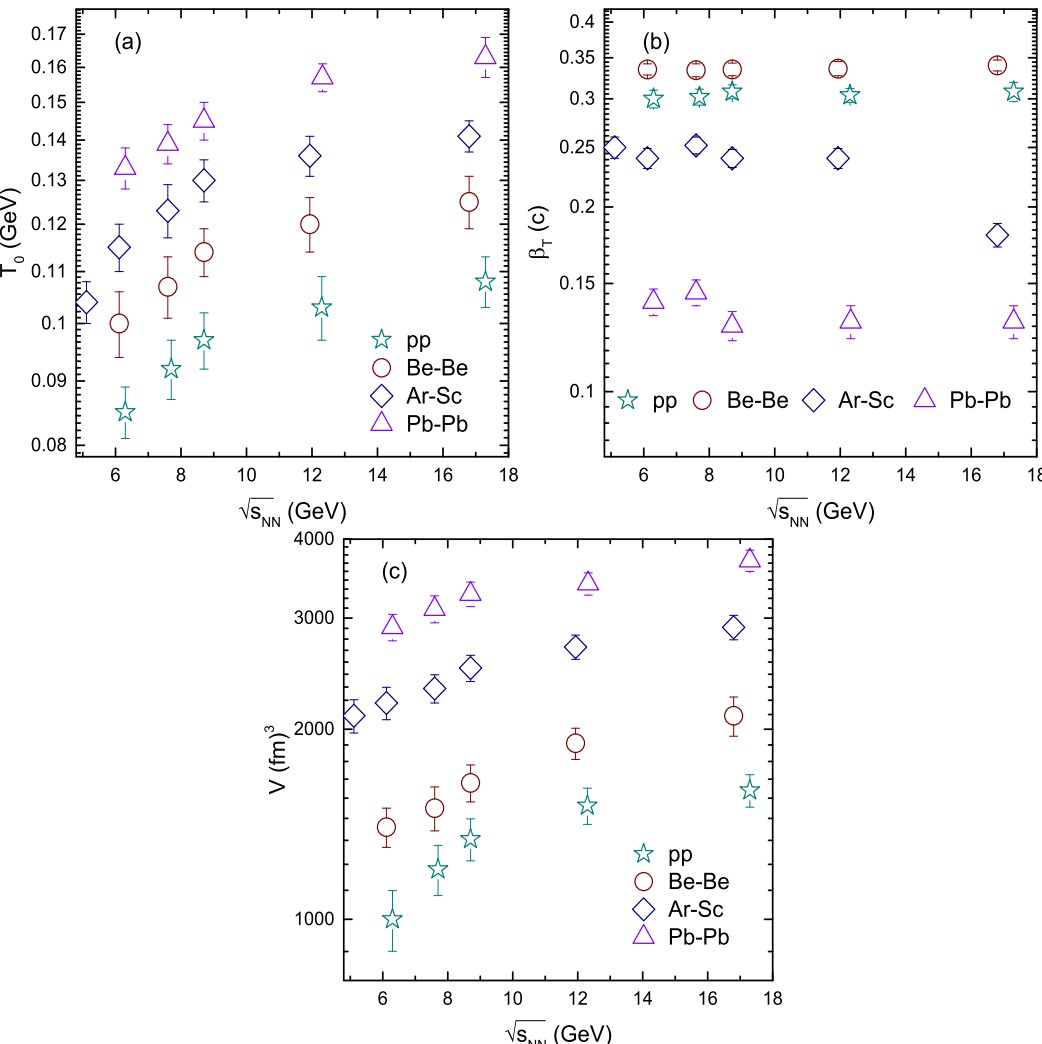

**Figure 2.** Dependence of (**a**) kinetic freeze-out temperature, (**b**) transverse flow velocity, and (**c**) kinetic freeze-out volume on collision energy and collision cross-section.

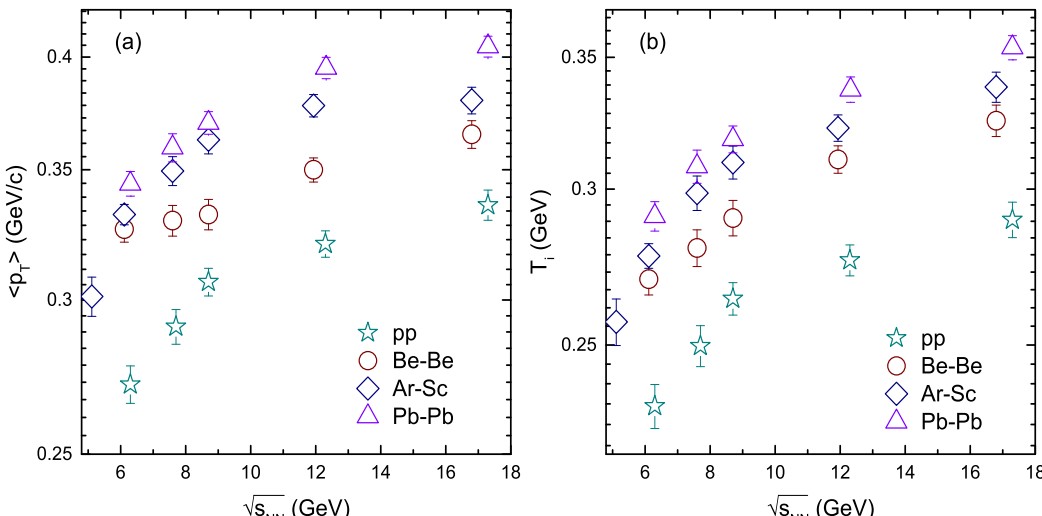

**Figure 3.** Dependence of (**a**) mean transverse momentum and (**b**) initial temperature on collision energy and collision cross-section.

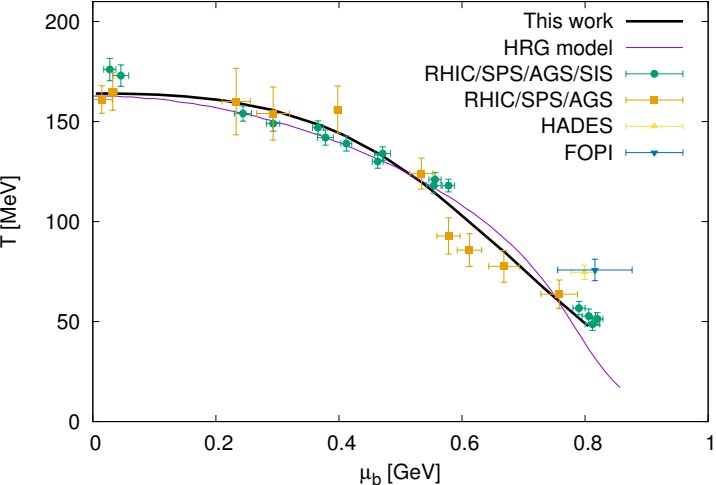

**Figure 4.** Dependence of the chemical freeze-out temperature on the baryon chemical potential, Equation (1) and Equation (2), respectively. The symbols represent various measurements. The thin curve stands for results obtained by the HRG model at constant higher-order moments of particle multiplicities. The curve label "This work" in the legend is determined by the statistical hadronization model fits.

## 4. Summary and Conclusions

We summarize here our main observations and conclusions as follows:

- The transverse momentum spectra of pions produced in inelastic proton–proton and most central Be–Be, Ar–Sc, and Pb–Pb collisions at different energies were studied by the Blast-wave model with Boltzmann–Gibbs statistics. The results are in agreement with the experimental data measured by the NA61/SHINE and NA49 experiments at SPS energies. In addition, we extracted the kinetic freeze-out temperature $T_0$, the initial temperature $T_i$, the transverse flow velocity $\beta_T$, and the kinetic freeze-out volume $V$.
- We studied the dependence of $T_0$ and $T_i$ on collision energy and cross-section. $T_0$ and $T_i$ were found to increase with the increase in collision energy and collision cross-section. This increase is due to the transfer of more energy in the system at higher energies and in large colliding systems.

- $\beta_T$ was observed to remain constant with the increase in energy due to the invariant collective flow with increasing energy. There is no dependence of transverse flow velocity on the collision cross-section.
- $V$ was found to increase with increasing collision energy because of the large initial bulk at higher energies, and it was also larger for large collision cross-sections. Furthermore, the mean transverse momentum increased with increasing collision energy due to a greater transfer of energy in the system at higher energies. It was also observed that mean transverse momentum was larger for the systems with large collision cross-sections because, in large collision section systems, the transfer of energy is larger.
- Finally, the chemical potential and temperature show excellent agreement in comparison to the HRG model and different experimental data at the chemical freeze-out, revealing the ability of the Blast-wave model with Boltzmann–Gibbs statistics to manifest features of the chemical freeze-out.

**Author Contributions:** Conceptualization, M.W., A.A.K.H.I. and M.A.; methodology, M.W., A.A.K.H.I. and A.A.; software, M.A., A.A.K.H.I. and A.A.; validation, M.A., M.W., A.N.T., S.J.L. and A.A.K.H.I.; formal analysis, A.N.T., S.J.L., H.Z.K. and Z.W.; investigation, A.A.K.H.I., Z.W., H.Z.K. and A.A.; resources, M.A. and A.A.; writing—original draft preparation, M.W., A.A.K.H.I. and M.W. All authors have read and agreed to the final published version of the manuscript.

**Funding:** This research was funded by Ajman University, Deanship of Research and Graduate Studies, Internal Research Grant No: (DGSR Ref. 2022-IRG-HBS-11).

**Institutional Review Board Statement:** Not applicable.

**Informed Consent Statement:** Not applicable.

**Data Availability Statement:** Not applicable.

**Acknowledgments:** This work was supported by Ajman University Internal Research Grant No. (DGSR Ref. 2022-IRG-HBS-11).

**Conflicts of Interest:** The authors declare no conflict of interest.

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
