# Peer review of "Dependence of Freeze-Out Parameters on Collision Energies and Cross-Sections"

_universe, doi:10.3390/universe9010044_

Round 1

Author Response

Kindly find our replies in attachment.

Reviewer 2 Report

This is an interesting study in the mainstream of model analysis of experiments creating extreme conditions of temperature and density. The main focus is the freezout, and its parameters. The main results in Figure 4 show a nice agreement with previous studies.

Before recommending publication, I would like to ask the authors to address the following points:

1. It is stated (pg.2)  that previous studies have contradictory results. It would be useful to have a concise account of the main points of contradiction.

2. It is mentioned that one could use different models. While arguments are given in favour of the choice of the Blast-Wave model, it is not clear why other modes should be dismissed. In particular, could that be that the contradictions reported in previous studies (point 1) are due to the model choice.

3. I understand that the bottom panels in Figure 1 should demonstrate the quality of the fits. It seems to me that there is a clear tension between data and fit, especially in the bottom panels. Some discussion would be desirable - for instance, would that change by using a different model? Lacking that, perhaps a word of caution would be appropriate.

Author Response

Kindly find our replies in attachment

Round 2

Author Response

Dear respected reviewer,

Please find our responses in attachment.

Best regards,
